# Architecture of a complete Bce-type antimicrobial peptide resistance module

Natasha L. George[1,2] & Benjamin J. Orlando [1] ✉

Gram-positive bacteria synthesize and secrete antimicrobial peptides that target the essential process of peptidoglycan synthesis. These antimicrobial peptides not only regulate the dynamics of microbial communities but are also of clinical importance as exemplified by peptides such as bacitracin, vancomycin, and daptomycin. Many gram-positive species have evolved specialized antimicrobial peptide sensing and resistance machinery known as Bce modules. These modules are membrane protein complexes formed by an unusual Bce-type ABC transporter interacting with a two-component system sensor histidine kinase. In this work, we provide the first structural insight into how the membrane protein components of these modules assemble into a functional complex. A cryo-EM structure of an entire Bce module revealed an unexpected mechanism of complex assembly, and extensive structural flexibility in the sensor histidine kinase. Structures of the complex in the presence of a non-hydrolysable ATP analog reveal how nucleotide binding primes the complex for subsequent activation. Accompanying biochemical data demonstrate how the individual membrane protein components of the complex exert functional control over one another to create a tightly regulated enzymatic system.

Antimicrobial peptides are chemically diverse molecules produced by a variety of organisms[1,2] that not only play an important role in regulating microbial community dynamics, but are also harnessed as clinical tools to treat bacterial infections. Many antimicrobial peptides that are active against gram-positive species share a common mechanism of action involving binding to lipid intermediates of the cell-wall synthesis pathway (lipid II cycle), which leads to inhibition of cell-wall synthesis and cell death (Fig. S1a)[3,4]. Examples of such peptides in clinical use include bacitracin, which is found in common over-the-counter antibacterial ointments for minor cuts and abrasions, and vancomycin, which is used to treat serious infections of the skin, blood, and bone.

In addition to producing diverse antimicrobial peptides, gram-positive organisms have also evolved mechanisms to evade attack from these agents[5]. One such mechanism widely found across Firmicute bacteria involves antimicrobial peptide sensing and resistance membrane protein complexes collectively known as Bce modules.

These modules are composed minimally of an ATP-binding cassette (ABC) transporter which defends against antimicrobial peptides without transport across the membrane, and an interacting two-component system containing a sensor histidine kinase and response regulator[6,7]. Of all such modules, the BceAB-RS system from *Bacillus subtilis*, after which the modules have been named, has been most extensively characterized[8–10]. In this module *bceA* encodes the ATPase domains, *bceB* encodes the membrane-spanning permease domain of the ABC transporter BceAB, and *bceS* encodes the histidine kinase that phosphorylates the response regulator encoded by *bceR* (Fig. 1a)[11]. Previous studies have established that BceAB mediates resistance against bacitracin, likely by freeing bacitracin from the lipid target undecaprenyl-pyrophosphate (UPP)[12,13]. Functional BceAB is required to activate the sensor kinase BceS through a proposed flux-sensing mechanism in which BceS senses the conformational cycling of BceAB, leading to BceS autophosphorylation and subsequent recruitment and phosphorylation of BceR[14,15]. Phosphorylated BceR then upregulates

[1]Dept. of Biochemistry and Molecular Biology, Michigan State University, East Lansing, MI, USA. [2]Dept. of Microbiology and Molecular Genetics, Michigan State University, East Lansing, MI, USA. ✉e-mail: orlandob@msu.edu

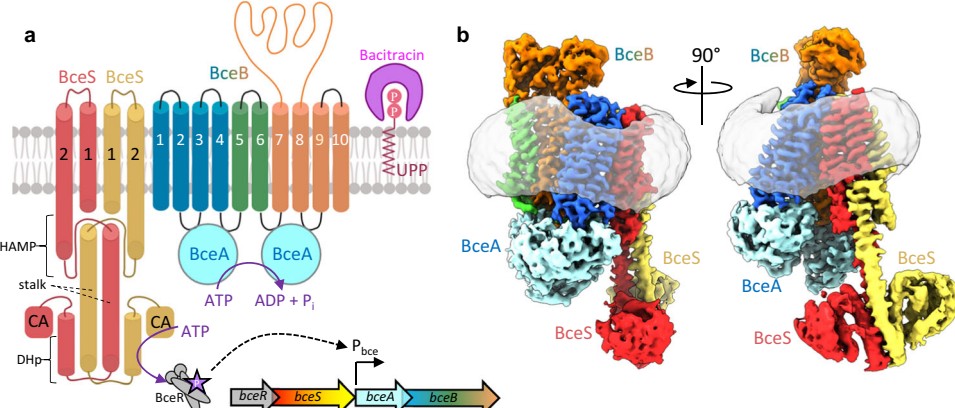

**Fig. 1 | Organization of the BceAB-S complex. a** Diagram showing the topology and stoichiometry of the BceAB-S membrane protein complex. The transmembrane (TM) helices of BceB are colored blue, green, and orange, and labeled with white numbers. BceS monomers are colored red and yellow with TM helices indicated with black numbers. Individual domains in the cytoplasmic region of BceS are labeled in black. Bacitracin binding to the lipid target UPP is shown as a pink crescent surrounding the phosphates of UPP. **b** Cryo-EM map of the BceAB-S membrane protein complex in a nucleotide-free state. Protein components are colored the same as in **a**. The detergent micelle is shown in grey surrounding the TM helices.

expression *bceA* and *bceB*, leading to a positive feedback loop that tunes the level of BceAB to meet the demand for bacitracin detoxification[15].

In recent work we used cryo-electron microscopy (cryo-EM) to provide the first two structural snapshots of a Bce-type ABC transporter, BceAB from *B. subtilis*[12]. While these studies revealed the structure and conformational changes of Bce-type ABC transporters, the architecture of a complete Bce module describing the stoichiometry and mode of interaction between the BceS kinase and BceAB transporter remained enigmatic. To understand this critical membrane protein interaction and the molecular details underpinning the Bce module flux-sensing mechanism, we have determined the cryo-EM structure of the complete BceAB-S complex from *B. subtilis* in nucleotide-free and ATPγS-bound states. Our analysis uncovers lipid mediated protein interaction between membrane components of a Bce module, a large degree of conformational heterogeneity within the sensor kinase, key conformational shifts that prime the complex for activation, and an unforeseen layer of reciprocal enzymatic regulation between membrane components of the complex.

## Results

### Cryo-EM structure of nucleotide-free BceAB-S

To investigate the assembly of membrane protein components in a Bce module, we overexpressed the genes encoding BceAB and the BceS sensor kinase from *B. subtilis* in *Escherichia coli* from a single plasmid containing a histidine tag on BceA (see methods). Following solubilization of bacterial membranes with lauryl-maltose neopentylglycol (LMNG) detergent, the BceAB-S complex was isolated by $Co^{3+}$-TALON affinity and size-exclusion chromatography. Comparison of the size-exclusion chromatogram with that obtained for the isolated BceAB ABC transporter revealed that co-production of BceS with BceAB resulted in a significant shift towards higher molecular weight species (Fig. S1b). SDS-PAGE analysis of the peak fractions from the two chromatograms confirmed that the molecular weight shift was a result of co-purification of BceS with BceAB (Fig. S1c). These results demonstrate that BceS forms a stable complex with BceAB, and that the entire membrane protein complex is stable in a detergent solubilized state.

To understand the molecular details of interaction between the BceS sensor kinase and BceAB, we pursued a cryo-EM structure of the detergent solubilized BceAB-S complex. Peak fractions from size-exclusion chromatography (Fig. S1b) were concentrated and applied to cryo-EM grids for high-resolution single-particle imaging. The

resultant micrographs show a consistent particle distribution in thin ice (Fig. S1d), and 2D class-averages from this dataset show features consistent with those seen previously for isolated BceAB along with additional transmembrane helices (TM) and soluble domains that extend below the detergent belt corresponding to BceS (Fig. S3a). The cytoplasmic dimerization and histidine phosphotransfer (DHp) and catalytic (CA) domains of BceS are very fuzzy or almost completely absent in many of the 2D averages, indicating a high degree of conformational heterogeneity. Despite this heterogeneity we reconstructed 3D volumes of the entire BceAB-S complex at sub-nanometer resolutions in two TM helix conformations (Fig. 1b and Figs. S2–S4). These initial 3D reconstructions show that the BceAB transporter is complexed with the BceS dimer to form an overall complex with five individual protein chains in a BceA:B:S ratio of 2:1:2 (Fig. 1a, b).

To gain more detailed insight into the interaction of BceS with BceAB, and a deeper understanding of the conformational heterogeneity present in the BceS cytoplasmic domains, we adopted segmented approaches to cryo-EM 3D classification and refinement focusing on specific regions of the complex. The results of these classification and reconstruction strategies are discussed below.

### Interactions between BceB and BceS TM helices

To understand how BceS interacts with BceB in the transmembrane region, we combined all particles that displayed clear secondary structure for TM helices in 3D classification and refined these particles into a single volume (Fig. S2). Residual signal subtraction focused on the TM region was followed with a round of 3D classification without alignment, generating two distinct classes of particles with high-resolution features in the TM region. Refinement of these two classes independently produced separate 3D volumes each at ~3.4 Å overall resolution (Figs. S2–S4). We refer to these two reconstructions as TM-state-1 and TM-state-2. The overall structure of the BceAB-S complex in TM-state-1 and TM-state-2 is nearly identical (overall RMSD ~ 0.6 Å), with very minor shifts in the position of TM helices (Movie S1). For simplicity we focus on TM state-1 for the following analysis.

Within the detergent belt 14 total TM helices were clearly identified, with 10 belonging to BceB and another 4 belonging to the BceS dimer. The cryo-EM maps were of sufficient quality to build atomic models of $TM_1$ and $TM_2$ of each BceS monomer de novo (Fig. S3d, e). The TM helices of BceS form a 4-helix bundle like that observed for the histidine kinase NarQ from *E. coli*[16] and the sensory rhodopsin transducer HtrII from *Natronobacterium pharaonis*[17] (Fig. S5a). The configuration of TM helices in BceS is inverted compared to NarQ and HtrII.

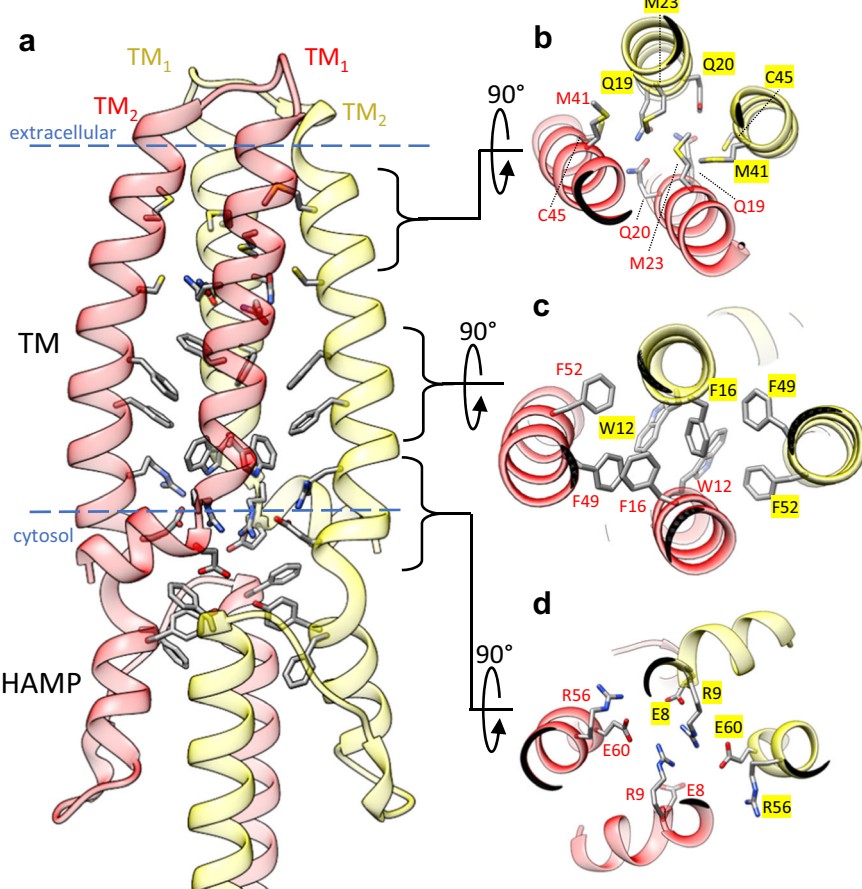

**Fig. 2 | BceS TM helix interactions. a** Overall configuration of the BceS TM helices and HAMP domain. Each monomer of BceS is colored either red or yellow. Blue dashed lines indicate the approximate boundary of the lipid bilayer. **b** View from the extracellular space of the tip of BceS TM helices. The core of the TM helix bundle contains a cluster of sulfur containing side-chains and hydrogen bonds between two pairs of glutamines. **c** View from the extracellular space of the middle of the BceS TM helix bundle, showing that the core of the bundle in the membrane interior contains a network of π-stacking interactions between aromatic side-chains. **d** View of the BceS TM helix bundle near the interface of the membrane and cytosol. An electrostatic network of charged residues mediates interaction between BceS TM helices.

The loop connecting $TM_1$ and $TM_2$ in BceS makes a left-handed turn such that $TM_2$ is positioned along the left-hand side of $TM_1$, whereas in NarQ and HtrII this loop makes a right-handed turn such that $TM_2$ is positioned along the right-hand side of $TM_1$ (Fig. S5a, b). A left-handed configuration such as that observed for BceS was previously identified in NMR structures of the isolated monomeric TM domain of *E. coli* QseC and ArcB[18].

Several types of interactions mediate dimerization of two BceS monomers (Fig. 2a–d). At the apex of the 4-helix BceS TM bundle near the extracellular space is a cluster of sulfur-containing residues including Met41 and Cys45. Located directly beneath this sulfur cluster, Gln19 and Gln20 from opposing BceS monomers form hydrogen bond interactions with one another (Fig. 2b). In the middle of the bilayer plane is a cluster of aromatic residues including Trp12, Phe16, Phe49, and Phe52 that form an extensive π-stacking interaction network (Fig. 2c). Directly below this π-stacking network and at the hydrophilic interface of the cytosolic leaflet of the plasma membrane is a cluster of arginine and glutamate residues including Glu8, Arg9, Arg56, and Glu60 (Fig. 2d) that balance the net charge within this area and form an intricate electrostatic interaction network between BceS monomers. Immediately beneath the TM helices of BceS in the cytosolic space is a helix-turn-helix HAMP domain formed by interactions between opposing BceS monomers (Fig. 2a). Within the core of the HAMP domain another cluster of aromatic residues including Phe63, Tyr64, and Phe86 from opposing BceS monomers interact through π-

stacking interactions (Fig. 2a). In total, our cryo-EM structures reveal that the intramembrane sensing histidine kinase BceS forms a classical 4-helix TM bundle and a HAMP transfer domain similar to those of other histidine kinases that contain extracellular ligand binding domains. We next focused our attention on how the BceS histidine kinase interacts with BceAB.

Although previous bacterial two-hybrid and pull-down assays demonstrated that BceS and BceB interact to form a membrane protein complex[11], the molecular details of interaction between these proteins remained obscure. A random chemical mutagenesis screen previously identified mutations in BceB that affected signaling through BceS[19]. Quite strikingly this study found several mutations in the C-terminal half of BceB ($TM_{8–10}$) which reduced signaling through BceS, and only two mutations in the N-terminal half of BceB ($TM_{1–4}$). From these studies and the conformational changes we previously observed in isolated BceAB[12], it would have been reasonable to predict that BceS interacts with the C-terminal half of BceB ($TM_{8–10}$). However, in our cryo-EM structures of BceAB-S we observe the TM bundle of BceS positioned near the N-terminal region ($TM_{1–4}$) of BceB (Fig. 3a, b). This positioning places BceS on the opposite side of BceB as a previously identified UPP binding pocket (Fig. 3b)[12].

Interestingly, the interaction between BceS and BceB in the TM region is limited to the very extracellular tip of the TM helices (Fig. 3a, b). The total buried surface area between BceB and the two BceS monomers is only ~460 Å² near the apex of BceB-$TM_3$, which

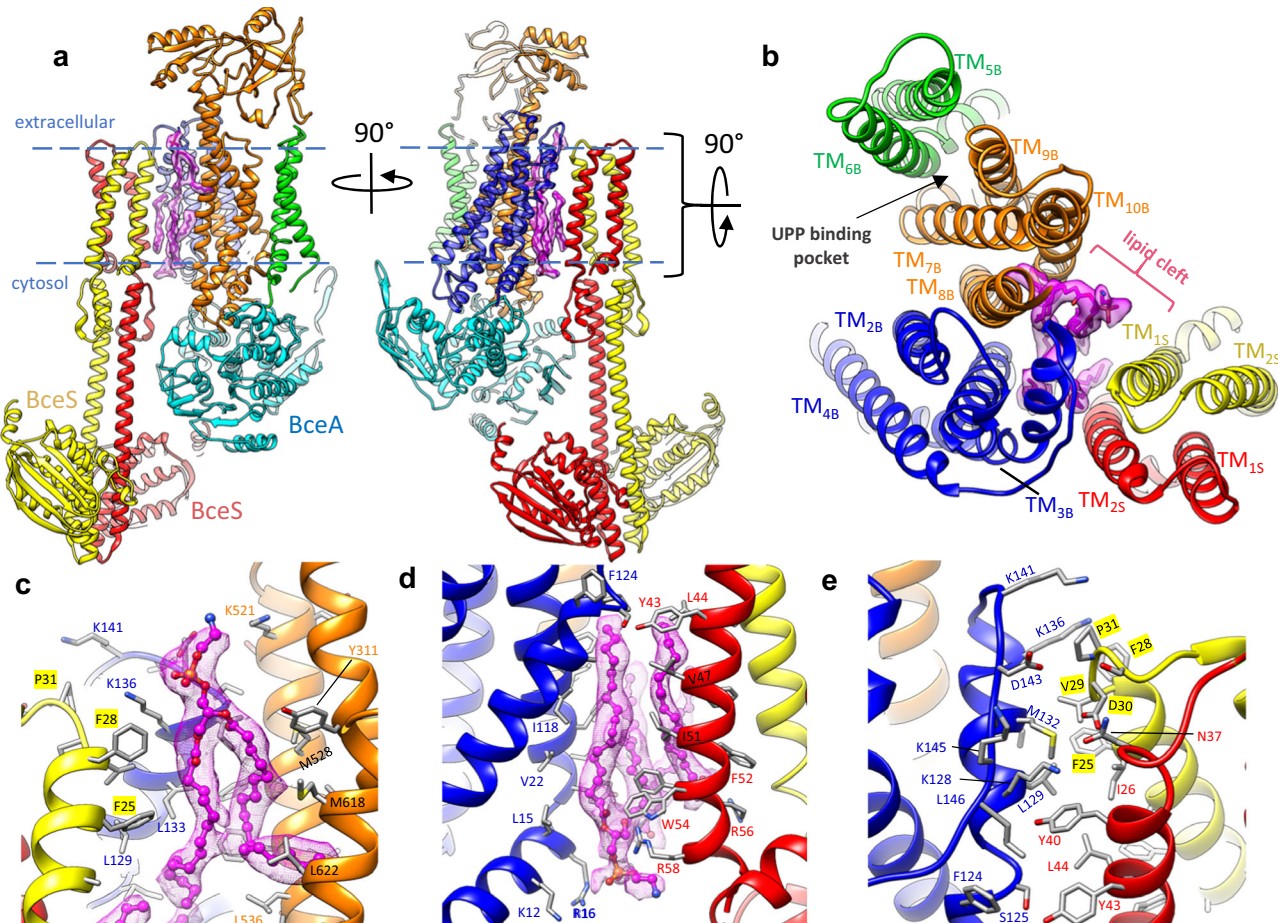

**Fig. 3 | Lipids mediate BceB-BceS interaction. a** Two rotated views of the atomic model of the BceAB-S complex. Individual protein chains and TM helices are colored the same as in Fig. 1. Purple sticks with surrounding transparent purple cryo-EM density indicate lipid molecules identified at the interface between BceB and BceS. Blue dashed lines indicate the approximate boundary of the lipid bilayer. **b** View from the extracellular space of the TM helix arrangement in the BceAB-S complex. Individual TM helices are labeled according to their number and whether they belong to BceB or BceS. Purple sticks with surrounding transparent purple cryo-EM density indicate lipid molecules that mediate interaction between BceB and BceS. The previously identified UPP binding pocket is indicated with a black arrow and is located on the opposite side of BceB as the site of BceS interaction. **c, d** Zoomed in view of lipid densities observed between BceB and BceS. Lipids are shown as magenta ball-and-stick with surrounding magenta mesh cryo-EM map. **e** Zoomed in view of the protein-protein interaction between BceB-TM₃ and BceS.

contacts TM₁ from one BceS monomer and TM₂ from the second BceS monomer (Fig. 3b, e). Although the buried protein surface area between BceB and BceS is relatively small, we identified several lipid-like densities between BceB and BceS TM helices within the bilayer plane (Fig. 3a–d). Unfortunately, the cryo-EM map is not of sufficient quality to unambiguously identify the specific lipids that fill the gap between BceB and BceS TM helices. Nevertheless, our cryo-EM structures clearly demonstrate that most of the interaction between BceB and BceS is mediated through protein-lipid interactions rather than direct protein–protein interaction.

### Conformational heterogeneity of BceS

Conformational flexibility of helical coiled-coils such as those found in histidine kinases is well-documented, and is proposed to underly interconversion between functional states through low energetic barriers[20,21]. Such conformational flexibility is apparent in 2D class-averages of the BceAB-S complex where the cytosolic region of BceS is often almost completely disordered (Fig. 4a), as well as in 3D classification of BceAB-S particles, where successive rounds of classification continually reveal multiple configurations of the BceS kinase (Fig. 4c). During 3D classification we routinely identified particle subsets that clearly contained all 4 BceS TM helices but showed few features for the cytosolic region of BceS below the detergent belt (Figs. S2, S6). These

results demonstrate that significant conformational flexibility in BceS begins at the HAMP domain and extends throughout the cytosolic DHp and CA domains.

We then applied 3D variability analysis (3DVA) in cryoSPARC to resolve the conformational heterogeneity in BceS, which revealed that the cytoplasmic region of BceS undergoes rotation around the axis of the stalk helices that connect the HAMP and DHp domains (Fig. 4b; component 1 and 2), and also translational shifts of the entire BceS cytoplasmic region relative to the BceAB transporter (Fig. 4b; component 2 and 3, and Movie S2). Much of the flexibility revealed in cryoSPARC 3DVA was also observed through standard 3D classification in Relion (Fig. 4c), which produced 3D classes demonstrating different conformations of BceS relative to BceAB. It is important to note that within these 3D classes the DHp and CA domains of BceS still remain poorly resolved compared to the TM regions, and subsequent 3D refinement produced reconstructions with largely disordered cytosolic regions of BceS.

Although conformational flexibility hampered near-atomic resolution reconstruction of the BceS soluble regions, we were able to separate two distinct classes of particles that reconstructed volumes showing most of the major secondary structural elements in the DHp and CA domains of BceS (Figs. 4d and S6b, d). These two volumes primarily differ in the angle of the BceS DHp/CA domains relative to

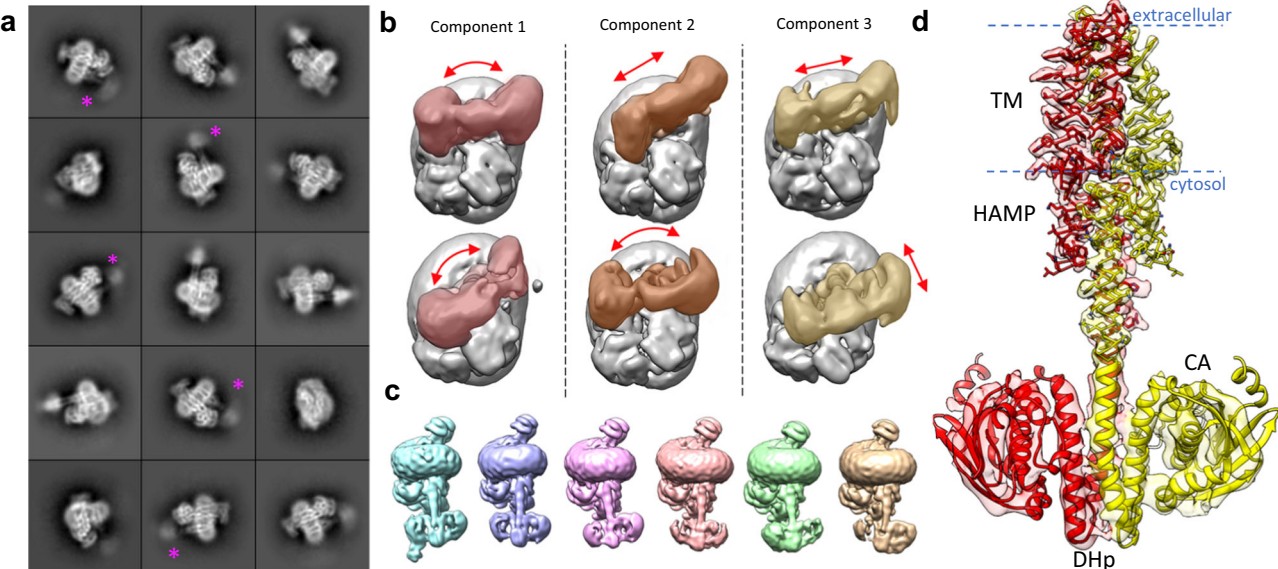

**Fig. 4 | BceS exhibits a high degree of intrinsic flexibility. a** 2D class-averages of detergent solubilized BceAB-S. Individual TM helices for BceS are present in the interior of the detergent micelle, while the cytoplasmic domains of BceS are often fuzzy. Purple asterisks indicate the BceS cytosolic domains in 2D averages where these domains exhibit a very large degree of structural heterogeneity. **b** The first three principal components of motion in the BceS cytosolic domains identified from 3DVA in cryoSPARC. The top and bottom rows indicate the extreme extent of motion observed in principal component. In each panel BceS is colored and BceAB and the detergent micelle are shown in grey. Red arrows indicate the direction of BceS motions observed. **c** Individual classes obtained from 3D classification of BceAB-S particles. All classes are oriented with the detergent micelle and BceAB transporter in the same position to highlight the variety of conformations that BceS can adopt within the complex. **d** View of the atomic model of BceS monomers fit into the local resolution filtered cryo-EM map of nucleotide-free BceAB-S. Near-atomic resolutions are obtained in the TM and HAMP domains, while the resolution steeply decreases entering the cytosolic DHp and CA domains due to inherent flexibility. Blue dashed lines indicate the approximate boundary of the lipid bilayer.

the rest of the complex (Fig. S6f). Although BceAB and the TM/HAMP region of BceS were resolved with side-chain resolution in these reconstructions, the cytosolic DHp and CA domains of BceS had a lower local resolution, indicative of the high degree of conformational variability (Figs. 4d and S6b, d). Despite the limited resolutions in the cytosolic region, the loop between helices in the DHp domain of BceS clearly takes a right-handed turn, placing the ATP binding site in the CA domain from one BceS monomer in proximity to His-124 of the opposite BceS monomer (Fig. S5c, d). Thus, BceS likely auto-phosphorylates His-124 in *trans*[20].

## ATP binding primes the BceAB-S complex for activation

Previous studies demonstrated that BceAB has complete control over BceS activation[14] and maintains the kinase in an inactive state in the absence of bacitracin[22]. Recognition of bacitracin and ATP binding/hydrolysis by BceAB are necessary to initiate signaling through BceS[9]. Based on these studies, we reasoned that ATP binding in the BceAB-S complex might drive conformational changes that prime the complex for activation upon subsequent bacitracin recognition. To investigate this hypothesis, we pursued a cryo-EM structure of wild-type BceAB-S with the non-hydrolysable ATP analog ATPγS. Initial 3D classification of the resultant dataset revealed two distinct particle populations, with one population showing a large degree of heterogeneity in the intracellular DHp and CA domains of BceS, and a second population in which the entire BceS kinase was more clearly resolved (Fig. S7a). Subsequent classification and refinement produced two cryo-EM maps of the ATPγS-bound BceAB-S complex, one of which was obtained at 3.1 Å overall resolution showing high resolution features BceAB and the TM and HAMP domains of BceS, but disordered BceS DHp and CA domains (Fig. S8b, d). The second reconstruction was obtained at a slightly lower overall resolution of 3.6 Å but shows features for the entirety of BceS (Fig. 5A and Fig. S8C, E).

In both reconstructions the binding of ATPγS (Fig. S8g) induces the BceA nucleotide binding domains (NBDs) to transition from the highly asymmetric configuration observed in the nucleotide-free state, into a conformation that closely resembles that seen in the isolated BceAB transporter bound to ATP (Figs. 5a and 6a, c). However, close inspection reveals that the BceA subunits are not fully dimerized and ATPγS engages the Walker-A motif and Tyr13 in the A-loop of BceA, but not the signature motif in the opposing BceA monomer (Fig. 6c). Thus, although ATPγS induced a more symmetrical orientation of BceA monomers within the BceAB-S complex, the monomers are not positioned to support ATP hydrolysis. More importantly, the conformational shifts induced by binding of ATPγS to the BceA subunits do not affect the overall conformation of BceB. Within the ATPγS-bound BceAB-S structure, the BceB TM helix configuration closely resembles that of the nucleotide-free state, rather than the closed conformation seen with isolated BceAB bound to ATP (Fig. 6a, b). Thus, it appears that in the BceAB-S complex, nucleotide binding induces a conformational shift in BceAB to an intermediate state residing between the nucleotide-free and fully collapsed ATP bound conformations.

In both reconstructions of BceAB-S bound to ATPγS the TM helices and HAMP domain of BceS remain unchanged relative to their configuration in the nucleotide-free state. However, in the reconstruction where the entirety of BceS is resolved, a distinct kink is observed around residue Gly96 in the stalk helix that connects the HAMP and DHp domains of BceS (Fig. 5a, b). The kink is only observed in the BceS monomer with the HAMP and stalk helix positioned closest to BceAB (Fig. 5a). Kinking of the stalk helix causes the DHp and CA domains of both BceS monomers to rotate ~30° towards BceA from their nucleotide-free positions (Fig. 5b). A morph between nucleotide-free and ATPγS-bound BceAB-S structures demonstrates that movement of BceA and BceS appears to be coordinated, with kinking of the BceS stalk helix coupled to ATPγS-induced movement of BceA subunits (Movie S3).

Although binding of ATPγS to BceA induced a kinked alternate conformation of the stalk, the overall configuration of the DHp and CA domains within the BceS dimer remain relatively unchanged. While the

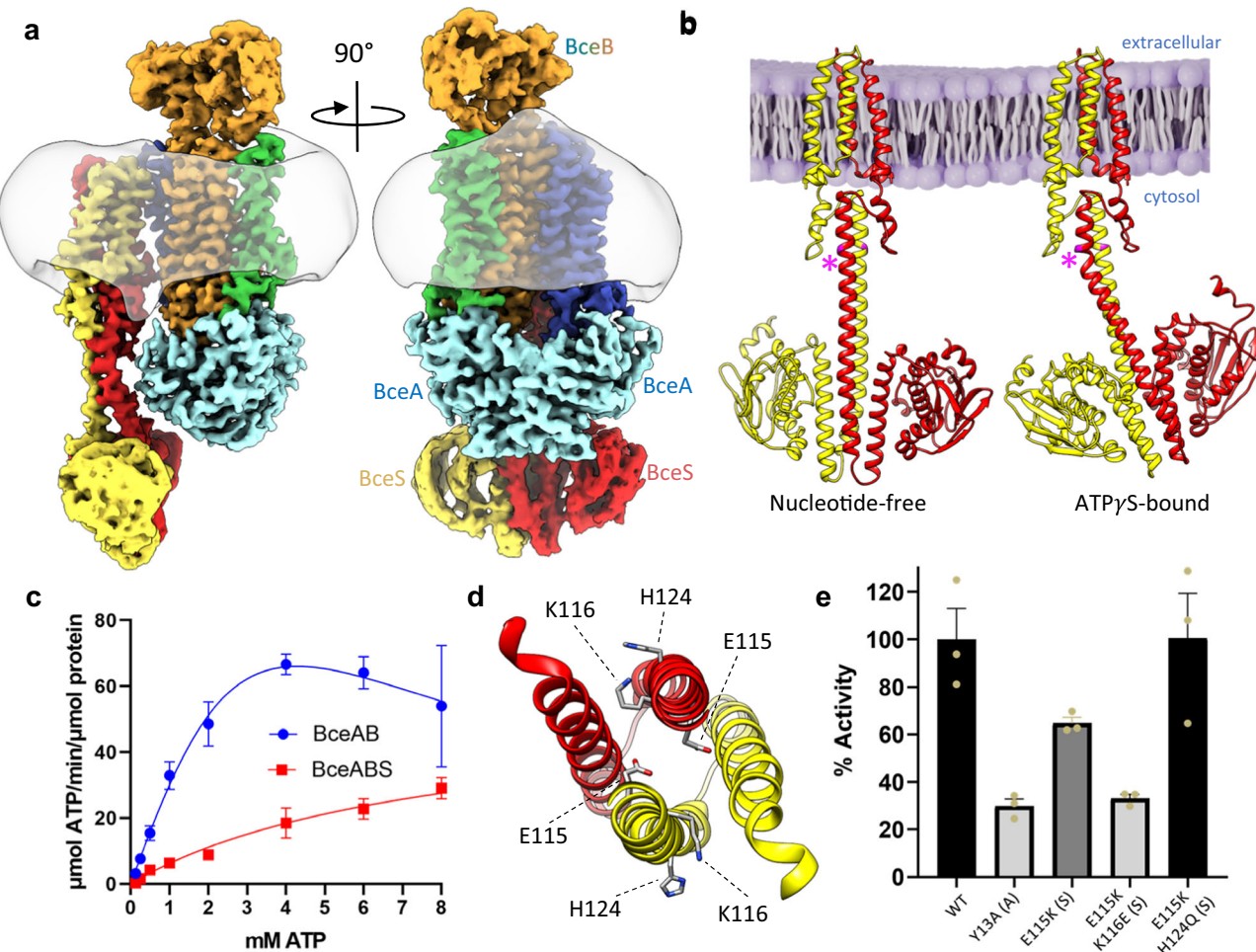

**Fig. 5 | ATPγS binding induces an intermediate conformation of BceAB-S. a** Two rotated views of the cryo-EM map of BceAB-S bound to ATPγS. Individual protein chains and TM helices are colored as indicated in Fig. 1. The detergent micelle is shown in transparent grey. Binding of ATPγS causes the BceA subunits (cyan) to collapse into a more tightly closed and symmetrical dimer. **b** Comparison of BceS configurations in nucleotide-free and ATPγS bound states of the BceAB-S complex. Binding of ATPγS to BceAB-S induces a ~30° kink in the stalk helix of one BceS monomer (pink asterisk). **c** ATPase measurements of detergent solubilized BceAB and BceAB-S complexes. Isolated BceAB exhibits higher basal ATPase activity than the complex with the BceS sensor kinase. Data points represent the mean across $n = 3$ triplicate measurements, and error bars represent standard deviation (SD) across the three measurements. **d** View of the DHp domain of BceS showing H124 that is auto-phosphorylated, and residues E115 and K116 that when substituted to charged-swap variants produce a constitutively active BceS in *B. subtilis*. **e** Comparison of maximal ATPase activity for WT and charge swap variants of the BceAB-S complex. Error bars represent standard error of the mean (SEM) across $n = 3$ triplicate measurements (shown in tan spheres).

limited resolution of the cryo-EM map within the BceS CA domains prevents identification of any bound ATPγS in this region, the catalytic domain is still positioned far from the phosphorylatable His124 in the DHp domain. Thus, further conformational change beyond that observed in our ATPγS bound BceAB-S structure would be required to form a Michaelis complex of BceS bound with nucleotide and poised for phosphate transfer to His124.

**BceS exerts enzymatic control over BceAB**

As mentioned above, previous studies have demonstrated that BceAB has exquisite control over BceS activation[14,15,22]. To investigate the possibility that BceS exerts reciprocal regulation on the basal ATPase activity of BceAB, we performed ATPase assays on detergent solubilized and purified preparations of isolated BceAB and BceAB-S complex. The BceAB-S complex displayed significantly reduced maximal ATPase activity compared to the isolated BceAB transporter (Fig. 5c). This result is consistent with our structural analyses demonstrating that nucleotide binding to BceA induces full closure of the NBD dimer in the isolated BceAB transporter[12], but not when the latter protein is in complex with BceS (Fig. 6c).

While the ATPase assay result in Fig. 5c demonstrates that BceS can throttle the basal ATPase activity of BceAB, the extent to which altered conformations or states of the BceS kinase contribute to this regulation remained unclear. Previous mutational analysis on BceS identified a critical bundle of electrostatic residues in the BceS DHp domain which can control the activation state of the kinase independent of the function of BceAB (Fig. 5d). In-vivo experiments in *B. subtilis* have shown that charge swap variants of Glu115 and/or Lys116 in BceS decouple BceS activation from signaling through BceAB and allow auto-phosphorylation of BceS in the absence of added bacitracin[22]. To probe the role of BceS activation on reciprocal enzymatic control of BceAB, we purified BceAB-S complexes containing different combinations of these charge swap variants (with or without additional mutation of His124 in BceS) and assessed their in-vitro ATPase activity (Fig. 5e). The E115K variant of BceS reduced the ATPase activity of the BceAB-S complex compared to WT protein, and a E115K/K116E double charge swap variant of BceS further reduced the ATPase activity to a level comparable to that seen with an ATP binding-deficient Y13A variant of BceA. Interestingly, addition of a H124Q substitution on top of E115K resulted in BceAB-S with

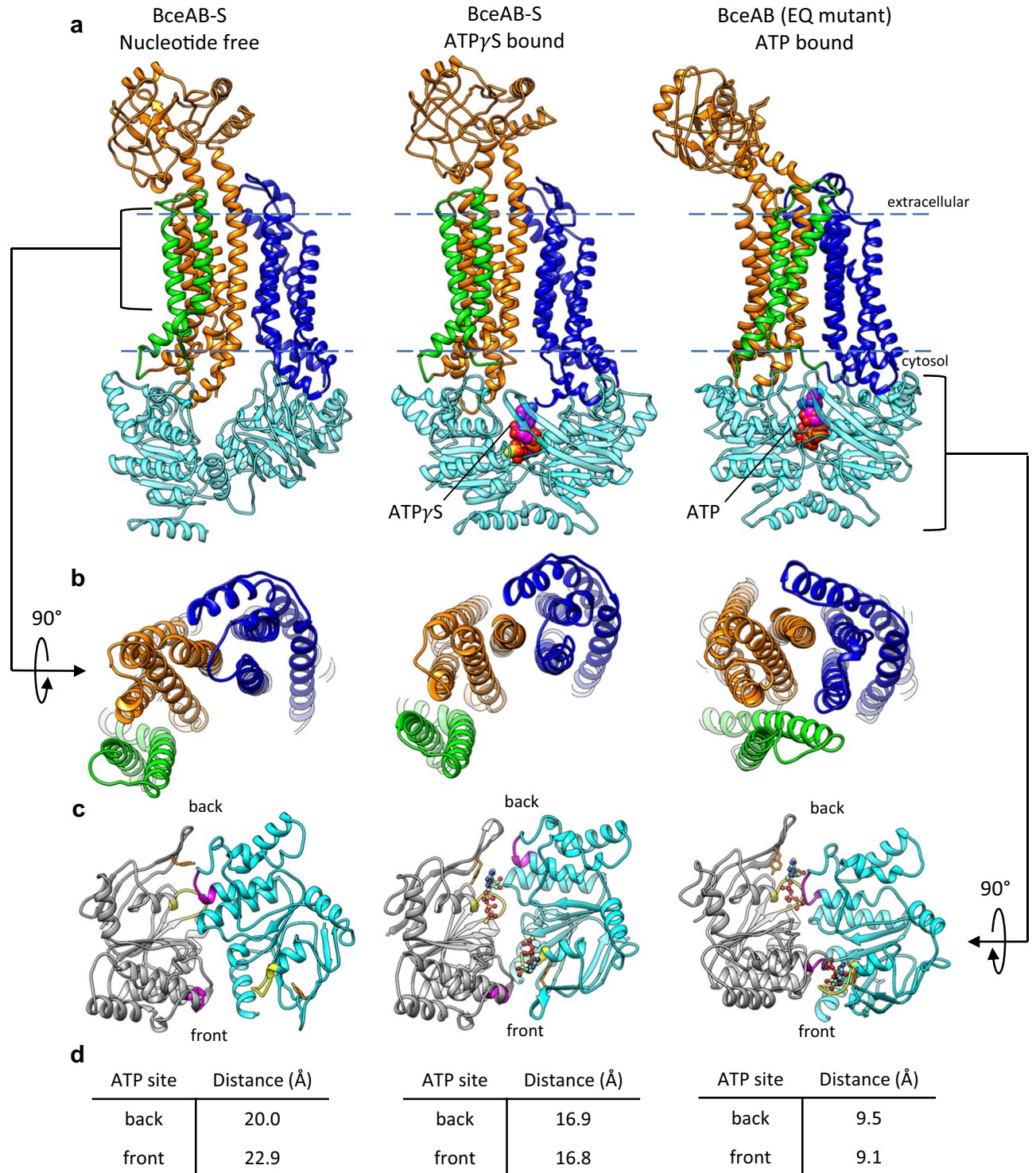

**Fig. 6 | Comparison of BceAB-S and BceAB Structures. a** Comparison of the conformations adopted by BceAB either in isolation or in complex with the BceS kinase, and in different nucleotide states. BceS is omitted for clarity. Whereas binding of ATP (magenta spheres) to isolated BceAB induces closure of the TM helices, binding of ATPγS (magenta spheres) to the whole BceAB-S complex does not induce a collapse of the BceB TM helices. Blue dashed lines indicate the approximate boundary of the lipid bilayer. **b** 90° rotated view relative to **a** showing the TM helix arrangement of BceB in each structure. **c** View of the BceA NBDs in the three structures. For clarity, one BceA monomer is colored grey and the other monomer is colored cyan. The Walker A motif is colored yellow, and the ABC signature motif is colored magenta. ATPγS and ATP are shown as ball-and-sticks. **d** Distance measured between residue 47 in the Walker A motif and 146 in the ABC signature motif for the two ATP binding sites in a dimer of BceA. The front and back sites are indicated in **c**.

ATPase activity comparable to the WT complex. While it would appear from this result that autophosphorylation of His124 in BceS could be a major determinant in reciprocal regulation of BceAB ATPase activity, analysis with Phos-tag SDS-PAGE acrylamide gels or fluorescent Phos-tag gel stains[23] have thus far failed to detect in-vitro phosphorylation of BceS with any of our WT or variant complexes of BceAB-S. Nevertheless, the results of ATPase assays collectively suggest that the activation state (and/or conformational landscape)

of BceS can have drastic effects on the ATPase activity of the BceAB transporter.

## Discussion

Bce modules pair an ABC transporter lacking membrane transport capability with a two-component signaling system lacking a dedicated sensing domain to provide many Firmicutes a first line of defense against antimicrobial peptides that target lipid intermediates of cell wall synthesis. These modules have been proposed to utilize a flux-sensing mechanism wherein conformational cycling of the ABC transporter activates the two-component system, resulting in upregulation of the transporter to keep up with the demand for antimicrobial peptide detoxification[15,22,24]. Our cryo-EM structures and biochemical data reveal how the individual components of a Bce module assemble into a functional membrane protein complex and begin to illustrate how conformational changes in the complex mediate a flux-sensing mechanism.

Our cryo-EM structures of BceAB-S demonstrate that the TM helices of the BceS sensor kinase interact with the first four TM helices of BceB (Fig. 1b), and that much of this interaction is mediated by membrane lipids (Fig. 3a, b). Previous biochemical studies demonstrated that activation of BceS by conformational cycling of BceAB is accompanied by piston-like displacements of $TM_2$ in BceS[22]. Although we do not observe any piston-like movement of BceS $TM_2$ in our various cryo-EM structures, it appears that the conformational coupling between the TM domains of BceB and BceS likely involves specific interactions with membrane lipids. Further structural and biochemical analysis of the BceAB-S complex in alternate conformational states will be required to unravel precisely how conformational cycling of BceAB initiates BceS activation, and the role that membrane lipids play in this overall process.

To our knowledge, our structures of BceAB-S represent the first cryo-EM structures of a full-length membrane-embedded histidine kinase. Unlike methods such as x-ray crystallography where individual domains are observed in isolation or full-length kinases are conformationally restricted in a crystal lattice, cryo-EM can provide insight into the conformational landscape of histidine kinases. Current models of histidine kinase function favor a statistical-thermodynamic model in which signal transduction through the kinase can be thought of as a series of thermodynamically linked equilibria between conformations of individual protein domains[20]. In this model the individual domains of a histidine kinase (TM, HAMP, DHp, CA, etc...) populate a landscape of thermodynamically linked independent conformational states and dynamics. This situation is readily apparent in our cryo-EM structures of BceAB-S, where in the resting state BceS samples a wide variety of conformations (Fig. 4a–c). Even in the states that can be resolved through 3D classification (Fig. 4c) significant conformational heterogeneity is still present in the DHp and CA domains, which severely limits resolution in these regions. Our observations of BceS conformational flexibility in cryo-EM reconstructions appears highly consistent with a statistical-thermodynamic model of histidine kinase function, with the individual domains of the sensor kinase adopting a variety of conformations relative to one another.

In addition to the conformational flexibility of BceS, our structures of the BceAB-S complex reveal conformational and functional differences between the full transporter-kinase complex and BceAB alone. In a previous study with isolated BceAB, ATP binding induced full closure of the BceA NBDs and a concomitant closure of the BceB TM helices and tilting of the extracellular domain[12]. In contrast, ATPγS binding within the full BceAB-S complex induced movement of the BceA subunits into a partially closed state unable to support ATP hydrolysis, without changes in the BceB TM helices (Fig. 6a–c). Not only do the BceAB and BceAB-S complexes adopt different conformations upon nucleotide binding, but the ATPase activity of each complex is significantly different (Fig. 5c). These results demonstrate

that BceAB and BceS reciprocally regulate one another, with BceAB being strictly required for BceS activation, and BceS throttling the intrinsically high basal ATPase activity of BceAB to prevent futile rounds of ATP hydrolysis in the absence of antimicrobial peptides.

Together with previous biochemical studies, our cryo-EM structures begin to paint a mechanistic picture of the overall signaling process through Bce-modules (Supplementary Fig. S9). In this model the BceAB-S complex in the nucleotide-free state (Supplementary Fig. S9; state 1) displays asymmetric BceA subunits, and highly flexible cytosolic regions of BceS. Subsequent binding of ATP pre-loads BceA with nucleotide and brings these domains into a more symmetric configuration, but one that is still insufficient for ATP hydrolysis. Along with movement of BceA in response to nucleotide binding, BceS adopts a kinked conformation but the DHp and catalytic domains remain in a conformation insufficient to support autophosphorylation of His124 (Supplementary Fig. S9; state 2). This overall configuration likely represents a sensing-ready state that would be encountered in *B. subtilis* prior to challenge with bacitracin. Once UPP-bacitracin complexes are encountered, ATP hydrolysis by BceAB would be stimulated, and in turn initiate signaling and autophosphorylation of BceS (Supplementary Fig. S9; state 3). Previous reports suggest that this signaling involves a piston-like movement of $TM_2$ in BceS[22], and our current structural analysis suggests that signal transduction through the entire complex may also involve interactions between BceA and BceS. While many of the molecular details underlying bacitracin recognition and signaling activation remain to be uncovered, this model provides a plausible explanation for the tight regulation of Bce module signaling and subsequent upregulation of BceAB to rapidly respond to antimicrobial peptide induced stress.

It is important to note that under conditions of antimicrobial peptide induced stress, both BceAB and BceAB-S complexes are present within the same cell. In this situation, the differing activity levels and conformational changes of these two complexes likely reflect distinct functions. BceAB alone is most abundant under bacitracin stress, and its high activity allows for rapid removal of the antimicrobial peptide from target lipids through ATP-driven changes in BceB. BceAB-S, however, is more tightly regulated for precise and highly tuned sensing of current stress levels

## Methods

### Protein expression and purification

The overlapped genes encoding BceA-BceB, and the gene encoding BceS were PCR amplified from *B. subtilis* genomic DNA and assembled into a petDUET-1 vector using NEB HiFi Assembly according to the manufacturer's protocol. This procedure generated a plasmid driving production of BceB and N-terminally 6x-His tagged BceA from one T7 promoter, and BceS driven from the second T7 promoter. The expression plasmid was transformed into *E. coli* C41(DE3) cells and a starter culture from a single colony was grown at 37 °C in Luria Broth (LB) medium with 100 μg/mL ampicillin. The starter culture was used to inoculate 6 L of Terrific Broth (TB) medium with 100 μg/mL ampicillin and Antifoam-204 in baffled Fernbach flasks. Cultures were grown at 37 °C to an optical density ($OD_{600}$) of ~0.8 before reducing the temperature to 18 °C and inducing protein expression with 0.4 mM IPTG. After ~16 h of induction the cultures were harvested by centrifugation and resuspended in lysis buffer (25 mM Tris (pH 8), 150 mM NaCl, 10% glycerol, 5 mM β-mercaptoethanol) supplemented with 1 μg/mL pepstatin A, 1 μg/mL leupeptin, 1 μg/mL aprotinin, 0.6 mM benzamidine, and ~5,000 units of Dr. Nuclease (Syd Labs).

Bacterial cells were lysed by sonication and membranes isolated by ultracentrifugation at ~100,000 g for 1 h. Isolated membranes were resuspended in lysis buffer and solubilized by stirring for 1 h at 4 °C with lauryl maltose neopentyl glycol (LMNG) added to a final concentration of 1%. Insoluble material was removed by ultracentrifugation at 100,000 g for 1 h and the resulting supernatant was applied to

Co³⁺-Talon resin. The resin was washed with ~10 column volumes of buffer A (25 mM Tris (pH 8), 150 mM NaCl, 10 mM imidazole, 10% glycerol, and 0.005% LMNG) before eluting bound protein with buffer A containing 250 mM imidazole. The eluted protein was concentrated in a 100 kDa MWCO spin concentrator and injected onto a Superdex 200 Increase 10/300 GL column equilibrated in 25 mM Tris (pH 8), 150 mM NaCl, and 0.005% LMNG. Peak fractions corresponding to the intact BceAB-S complex were pooled, concentrated in a 100 kDa MWCO spin concentrator and used immediately for cryo-EM studies, or flash frozen in liquid nitrogen and stored at −80 °C for later use.

## ATPase assay
Site-directed mutagenesis of the *bceA* and *bceS* genes was performed using a Q5 mutagenesis kit from NEB and standard manufacturers protocols (oligonucleotide primer sequences used in this study are included in the accompanying Source Data file). Plasmids encoding variants of BceAB-S were verified through Sanger sequencing at the RTSF Genomics Core at Michigan State University. Colorimetric ATPase assays were modified from a previously described protocol[12] and performed in 96 well plates. To each well 2 μg of purified BceAB or 3 μg BceAB-S was added, followed by buffer containing the indicated concentration of ATP and MgCl₂ to create a total reaction volume of 50 μL. Samples were incubated at 37 °C for 30 min before stopping the reaction by addition of 50 μL of a 12% (w/v) SDS solution. Then 100 μL of a 1:1 mixture of 12% ascorbic acid (w/v) and 2% ammonium molybdate (w/v) was added, followed by 150 μL of a solution containing 2% sodium citrate (w/v), 2% sodium (meta)arsenite (w/v), and 2% acetic acid (v/v). Absorbance at 850 nm was measured in a Molecular Devices ID5 plate reader and converted to total ATP consumed using a standard curve generated with $K_2HPO_4$ solutions. The rate of ATP consumption (μmol ATP/min/μmol protein) was plotted in GraphPad Prism and curves were fit with a model of substrate inhibition Velocity = $V_{max}$*[ATP]/($K_m$ + [ATP]*(1 + [ATP]/$K_i$) or a standard Michaelis Menten equation Velocity = Vmax * [ATP]/($K_m$ + [ATP]).

## Cryo-EM imaging and data processing
Grids for cryo-EM imaging were prepared on a Vitrobot Mark IV by applying 2.5 μL of purified BceAB-S at ~7 mg/mL to Quantifoil R2/2 200 mesh grids that had been glow-discharged for 45 s at 15 mA in a Pelco EasyGlow. In the case of ATPγS bound BceAB-S the purified protein complex was incubated on ice with 5 mM ATPγS and 5 mM MgCl₂ for 30 min before applying samples to cryo-EM grids. The grids were blotted for 5 s at 4 °C, 100% humidity, and a blot-force of 1 before being plunge frozen in liquid ethane cooled by liquid nitrogen. Frozen grids were screened for ice quality and particle distribution using EPU software on a Talos Arctica equipped with a Falcon-3 detector at the RTSF Cryo-Electron Microscopy Facility at Michigan State University. Final data collection was performed at Purdue University using Leginon[25] on a Titan Krios with a K3 direct electron detector and Gatan Quantum GIF energy filter set to a 20 eV slit width. Movies were collected in counting mode with a pixel size of 0.872 Å and a total dose of 60.5 electrons/Å².

Movies were corrected for beam-induced motion by performing patch motion correction in cryoSPARC[26], and CTF parameters were determined by patch CTF estimation in cryoSPARC. Micrographs with CTF fit parameters worse than 6 Å were discarded, and further manual inspection was performed to remove obviously poor micrographs. Following particle picking and several rounds of 2D classification in cryoSPARC to remove bad particles, initial models for 3D classification were constructed using ab initio reconstruction in cryoSPARC. Where indicated the particles were transferred to RELION 4.0[27] for further 3D classification and signal subtraction, before finally being reimported to cryoSPARC for a final round of non-uniform refinement[28]. Resolutions of all maps were calculated using the gold-standard FSC and local resolutions were calculated in cryoSPARC.

## Model building and refinement
To build atomic models into the cryo-EM maps for the nucleotide-free BceAB-S complex, we initially rigid-body docked the previous nucleotide-free BceAB transporter structure (PDB = 7TCG) and AlphaFold[29] predicted models of *B. subtilis* BceS monomers into the cryo-EM map using UCSF Chimera[30]. These models were manually adjusted in COOT[31] and refined in real-space using phenix.real_space_refine in the PHENIX software suite[32]. Iterative rounds of real-space refinement in PHENIX and manual model adjustment in COOT were performed to optimize the overall model properties and fit to the experimental cryo-EM map.

To build atomic models of the BceAB-S complex in the ATPγS bound state, the refined nucleotide-free BceAB-S structure was first rigid body fit into the cryo-EM map as a whole using the BceB and BceS TM helices as an initial guide. Rigid body fitting of individual BceA protein chains and the cytoplasmic regions of BceS was then performed in UCSF Chimera to adjust for shifts induced by ATPγS binding. The resulting model then underwent iterative rounds of real-space refinement and manual adjustment in PHENIX and COOT as described above for the nucleotide-free complex. The initial rounds of real-space refinement in PHENIX utilized the morphing option to account for large secondary structural shifts that were not resolved through initial rigid-body docking. All models were assessed for appropriate stereochemical properties and fit to the experimental cryo-EM map using MolProbity[33]. Figures were created using UCSF Chimera, UCSF ChimeraX[34], and BioRender.com.

## Reporting summary
Further information on research design is available in the Nature Portfolio Reporting Summary linked to this article.

## Data availability
Atomic coordinates and associated electron microscopy maps for the six structures reported in this publication have been deposited in the Protein Data Bank (PDB) and Electron Microscopy Data Bank (EMDB) under the following accession numbers: Nucleotide-free BceAB-S TM State 1 (PDB 8G3A, EMDB 29690), Nucleotide-free BceAB-S TM State 2 (PDB 8G3B, EMDB 29691), Nucleotide-free BceAB-S BceS State 1 (PDB 8G3F, EMDB 29694), Nucleotide-free BceAB-S BceS State 2 (PDB 8G3L, EMDB 29701), ATPγS bound BceAB-S High-res TM (PDB 8G4C, EMDB 29716), ATPγS bound BceAB-S Kinked BceS (PDB 8G4D, EMDB 29717). The initial model of BceAB used as a reference for model building is available at the PDB under the following accession number (PDB 7TCG). The AlphaFold model of BceS used as an initial template for model building is available at the AlphaFold Protein Structure Database [https://alphafold.ebi.ac.uk/entry/O35044]. Source data are provided with this paper.

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

## Acknowledgements

Research reported in this publication was supported by the National Institute of General Medical Sciences of the National Institutes of Health under Award Number R35GM146721 to B.J.O. The content is solely the responsibility of the authors and does not necessarily represent the official views of the National Institutes of Health. Electron Microscope support at Purdue University was established under the NIH Midwest Cryo-EM Consortium grant number 1U24GM116789-01A. We would like to thank Dr. Sundharraman Subramanian for help with electron microscope operation at the RTSF Cryo-Electron Microscopy Core at Michigan State University. We are also grateful for the NIH Midwest Cryo-EM Consortium and Dr. Thomas Klose, Dr. Frank Vago, and Dr. Wen Jiang for providing microscope time and assistance with data collection at Purdue University. We would also like to thank all members of the Orlando laboratory and Dr. Robert Hausinger and Dr. Sean Crosson for careful reading of the manuscript.

## Author contributions

N.L.G. performed molecular cloning, protein purification, cryo-EM imaging, data processing, and enzymatic/biochemical analysis. B.J.O. assisted with cryo-EM data processing and model building. Both authors analyzed the data and wrote the manuscript.

## Competing interests

The authors declare no competing interests.
