## [Peer Review File · Nature Communications]

Reviewer comments, first round

Reviewer #1 (Remarks to the Author):

In this manuscript, George and Orlando describe the structure of the antimicrobial peptide Bce ABC transporter in complex with the histidine kinase BceS. This is a major accomplishment, that brings to light many exiting questions on this emerging field (emerging on the structural level). No doubt that this work will serve as a reference for future works in the field and will be highly cited. The authors have done a remarkable work of purification/biochemistry/enzymology, and a vast amount of work on cryoEM data analysis, truly state-of-the-art. Overall, this work brings to light the complex, and the variability of the complex, which plays a role in its regulation. There are many unanswered questions on how the histidine kinase manages the autophosphorylation and signal transduction, which is without a doubt the future directions of research and I look forward to those results. For the present work, I have 1 major comment, and minor comments. After answering the questions, I recommend this work for publication.

Major comment:

Figure 3: the authors describe the interaction between BceS and BceB, and identify densities in-between the TM helices that would mediate interaction. Do the authors have evidence that there is lipid still present in the sample after the purification? A lipidomic test could answer the question. Otherwise, I would argue for modelling of LMNG instead, that could easily be mistaken for lipids. I can't see from the picture if the authors modelled UPP or phospholipids, and also the UPP binding site previously identified in the PNAS paper of the authors is empty this time, which to me looks like it is most likely LMNG in the densities. The text would need to be modified accordingly.

Minor comments:

The authors solved the structure of the complex solubilized in LMNG and use the word detergent "micelle" to describe the solvent seen around the hydrophobic region in the cryoEM reconstructions. I encourage the authors to use the word "belt" instead of micelle, as the micelle is a different object. It has been shown that LMNG micelles are long rod-shaped objects of around 400-600 monomers (aggregation number). With detergent quantification, we know that there are about 150 LMNG monomers around a 12TM type-IV ABC transporter, so I imagine there might be around 200 or less around the Bce complex. It is thus a different object than the micelle.

Figure S5: The authors mention a different orientation of TM helices of BceS. They have done an extensive analysis of cryoEM data, truly state-of-the-art, and a nice work to reach as high resolution as possible while looking at flexibility. The density maps look really good, and the model placing looks convincing. However, even at this resolution, it is still possible to make a mistake in the modelling so any help can be a good. I wonder about the QseC and ArcB, if by flipping the view they would look like NarQ ones, thus making BceS different on its own. Can the authors comment on alphafold predictions of BceS, and see if the helices are modelled the same way as they did. If yes, then it's another point that this is correct. If no, it would be interesting to see the reverse modelling to judge what it looks like.

"This result is consistent with our structural analyses demonstrating that nucleotide binding to BceA induces full closure of the NBD dimer in the isolated BceAB transporter¹², but not when the latter protein is in complex with BceS (Suppl. Fig. S9)." I believe the authors want to refer to SFigure 8 instead, I don't have a SFigure 9.

Figure 4 shows a beautiful example of protein flexibility sampled and analyzed by cryoEM, revealed by 3DVA analysis. The movements shown by this analysis (and also using 3D classification) are very intriguing and ask for many questions on the role of this variability on the kinase

autophosphorylation function, to be addressed in further studies I hope. It would be useful to measure these movements (rotation, translation, etc...) more accurately to analyze protein flexibility in terms of mechanics. Not required for this work, but perhaps in later studies.

Reviewer #2 (Remarks to the Author):

George, et al. report the cryoEM structure of the BceAB-S complex from *B. subtilis* in both the apo and ATPγS-bound states. The structure reveals how an unusual ABC transporter, BceAB, interacts with BceS, a sensory histidine kinase. BceAB is quite unique among ABC transporters; though most similar to MacB and LolCDE, it differs substantially in its extracytoplasmic domains. The authors show that BceS binds to BceAB via the transmembrane region, and suggest that BceS activation may be triggered by conformational changes in the ABC transporter during the process of bacitracin detoxification. Comparing the structure of BceAB-S in the apo and ATPγS states reveals a kinked conformation of BceS, which may be involved in communication between the ABC transporter and the kinase domain. This communication was probed with a small number of mutations in an ATPase activity assay.

So overall, the cryoEM structures appear to be well-determined and well-refined. The structures presented are intriguing and provide our first insights into how this unprecedented combination of an ABC transporter and sensory histidine kinase work together to sense antibiotics targeting the cell wall. We think this work will be of interest to several scientific communities, including researchers interested in ABC transporters, cryoEM of membrane proteins, antibiotic resistance, and bacterial mechanisms of environmental sensing and response. While we include some questions and suggestions for revision below, overall we thought this was a very exciting story and have only relatively minor criticisms.

MAJOR COMMENTS:

No Major Comments

MINOR COMMENTS:

Did the authors state what "bce" stands for? We couldn't find it.

It would be nice to show a model figure for how the authors think the whole system functions (either at the beginning by way of introduction, or at the end to summarize).

It would be helpful to add lines or shading indicating where the membrane region is on all figures (as in Fig. 2); this is missing in several places (e.g., Fig. 3). Also labels for cytoplasm, extracellular, etc.

Line 114: "two TM helix states": This wording is a little confusing to me. I take it to mean two different conformations that differ in the transmembrane region, but I am not completely sure this is correct.

Line 115: "two monomers": this makes me picture the BceS protomers as binding independently to BceAB, but it looks like BceS forms a dimer. Say that more clearly?

For readers less familiar with histidine kinases, it would be helpful and enlightening to compare the structure of BceS with other available structures. e.g., are the kinase domains similarly oriented? Are stalk lengths similar? etc.

Fig. 3A,B: The lipid mediated interaction between BceAB and BceS is quite interesting. However, the details of how this interaction is stabilized is not particularly clear in Fig. 3. Also, what sorts of direct protein-protein interactions are made at this interface? It might be helpful to add additional panels to this figure showing how each protein interacts with the bridging lipid and how BceS interacts with BceB.

Fig. 4B: The movie showing the 3DVA is nice, and Fig. S6F nicely illustrates one kind of motion in BceS. In contrast, it is a bit more difficult to see what is moving relative to what in Fig. 4B. Could these motions be shown in a different way (perhaps like the model-based representation in Fig. S6F)?

Line 268: refers to Suppl. Fig. S9, but I don't see that figure in the reviewer copy.

In a couple of places, the authors refer to "tilted BceS". Is this the same as "kinked"? If so, using the same nomenclature throughout would be more clear.

Fig 6A: ATP not labeled or mentioned in figure legend.

Table S1: Include model-map CC values.

Line 135-143: This seems to me like a surprising observation that would be interesting to comment on further. Is it possible that some of the structures are incorrect? Are these regions thought to derive from a common ancestor for both the left and right handed turns? In addition to the turn, it looks like the helix packing angles / superhelix handedness might also differ. Is this convergent evolution? Because the different connectivity between structures is surprising, it would be good to show density for the BceS TM1-TM2 linkers in Fig. S5 (one could imagine showing the linker density for NarQ and HtrII as well, for completeness).

Line 252-258: Might the kinking and movement of BceS be caused by nucleotide binding to the CA domains rather than nucleotide binding to BceA? If the interactions between the BceS and BceB TM regions don't change much, how would the nucleotide state of BceA be transmitted to BceS to cause the observed conformational change? This part of the mechanism is not clear to us, though perhaps it remains to be unraveled.

Regarding the ATPase Assay, does BceS have ATPase activity on its own? We were a little uncertain about how to interpret the results of the ATPase assays in light of the possibility (?) that BceA and BceS may both be capable of hydrolyzing ATP, and could also potentially allosterically regulate each other's activity.

The differences between the previous structure of ATP-bound E-to-Q mutant of BceAB, vs the current structure of BceAB-S bound to ATP-gamma-S are generally interpreted in terms of differences between BceAB in isolation and BceAB bound to BceS. But sometimes different strategies to stabilize the ATP bound state can give differing results. In addition, given the presence of multiple non-equivalent ATP binding sites (two different sites in BceA, plus 2 sites in BceS), the situation is extra complicated. I might suggest being a little more conservative in how these structural differences are interpreted, or perhaps mutants could be made to probe the role of ATP-binding at various sites in BceA vs BceS?

Reviewer #3 (Remarks to the Author):

The manuscript by Natasha George and Benjamin Orlando describes novel structures of the *B. subtilis* BceAB-S complex, implicated in antimicrobial peptide sensing (such as bacitracin). The manuscript builds on previous studies by the same group on the BceAB complex alone, which is an ABC transporter system evolved to sense, rather than transport, certain ligands. The transporter interacts with a two component system containing a His kinase (BceS) and response regulator. The current manuscript describes cryo EM structures of the BceAB transporter in complex with a BceS dimer. Somehow, bacitracin binding by BceAB activates BceS, leading to autophosphorylation of BceS' DHp domain. Although compromised by conformationally flexible particles, the new data describe apo and ATPγS-bound structures of the BceAB-S complex and reveal conformational changes of BceS that may be important for signal transduction.

Major points

- For a general reader, it would be useful to illustrate the signal process as a cartoon in Fig. 1.
- Fig. 3 – interaction of BceAB with BceS: The interaction of the BceS dimer with BceAB is not described in detail. Perhaps this is due to limiting resolution of the interface. However, are there any conserved residues that cluster at the interface? Considering the small interface, it is surprising that a stable complex is formed. In addition, the density in the lipid cleft could be shown in more detail (perhaps as a supplemental figure) to support the interpretation as a lipid molecule. Could it be a detergent molecule?
- Fig. 3 – BceS – BceA interaction: I understand that BceS' DHP and CA domains are insufficiently resolved in most maps. However, Fig. 3A shows a complete model that suggests potential interactions of BceA with one BceS protomer. Could the authors speculate on whether or not this interaction is significant? Are there any conserved residues at the interface that may transfer conformational changes of BceA to BceS?
- ATPgS-bound state of BceAB: This structure reveals a closer association of BceA's nucleotide binding domains in the presence of ATPgS. Yet, as stated, the NBDs do not close completely. This is in contrast to a previously obtained structure of a Walker-B mutant of BceAB bound to ATP. Thus, is it possible that the new structure is not completely closed due to ATPgS? Have the authors tried to stabilize the ATP-bound BceAB-S complex by mutagenesis, as described in Ref. 12? This seems important because the current structure would suggest that perhaps bacitracin sensing, i.e. changes within the TM region, are necessary for complete NBD closure.
- Fig. 5 – bending of BceS: The ATPgS-induced bending of BceS is interesting, however, it is not described how the kink is induced. What conformational changes of BcsAB reposition the BceS protomer?
- Fig. 5 - ATPase assays: It would be very interesting to compare the hydrolytic activity of BceAB w/o BceS in the presence and absence of bacitracin/UPP.
- Discussion: The discussion lacks a (speculative) description of the overall signaling process. How does bacitracin sensing lead to BceS autophosphorylation? What kind of conformational changes of BceAB are expected that could activate BceS? Where does bacitracin/UPP bind and would that interfere with BceS binding?

Minor points

The main text includes detailed experimental information that could be moved to the supplement, should space be limiting.

[redacted]

We are very pleased that three reviewers had overall positive comments. After analyzing the reviewer's comments and making the suggested corrections, we believe that our manuscript has been significantly strengthened. Highlighted below are point-by-point responses to the critiques raised by each reviewer. The original comments and suggestions from the reviewers are highlighted in *italicized and indented text*. Our responses and any corrective actions that have been taken with the manuscript are **highlighted in bold beneath each reviewer critique**.

Reviewer #1 (Remarks to the Author):

In this manuscript, George and Orlando describe the structure of the antimicrobial peptide Bce ABC transporter in complex with the histidine kinase BceS. This is a major accomplishment, that brings to light many existing questions on this emerging field (emerging on the structural level). No doubt that this work will serve as a reference for future works in the field and will be highly cited.

The authors have done a remarkable work of purification/biochemistry/enzymology, and a vast amount of work on cryoEM data analysis, truly state-of-the-art. Overall, this work brings to light the complex, and the variability of the complex, which plays a role in its regulation.

There are many unanswered questions on how the histidine kinase manages the autophosphorylation and signal transduction, which is without a doubt the future directions of research and I look forward to those results.

For the present work, I have 1 major comment, and minor comments. After answering the questions, I recommend this work for publication.

We thank the reviewer for their overall extremely positive comments. Undoubtedly, there is much more to learn about these complex systems, but we hope our current studies will serve as a reference for future works in the field as the reviewer suggests.

Major comment:

Figure 3: the authors describe the interaction between BceS and BceB, and identify densities in-between the TM helices that would mediate interaction. Do the authors have evidence that there is lipid still present in the sample after the purification? A lipidomic test could answer the question. Otherwise, I would argue for modelling of LMNG instead, that could easily be mistaken for lipids. I can't see from the picture if the authors modelled UPP or phospholipids, and also the UPP binding site previously identified in the PNAS paper

of the authors is empty this time, which to me looks like it is most likely LMNG in the densities. The text would need to be modified accordingly.

In an effort to identify potential lipids that fill the space between BceS and BceAB, we did indeed perform lipidomic profiling with LC-MS on purified BceAB-S and BceAB prior to atomic model building. These experiments were designed in an attempt to identify potential lipids that co-purify only with BceAB-S, or in significantly elevated proportions relative to BceAB. Preliminary results of these lipidomic profiling studies are shown below. As can be seen in the chart, significant amounts of membrane phospholipids co-purify with detergent solubilized BceAB and BceAB-S through affinity and size-exclusion chromatography. The prominent lipid species identified in this approach were mostly common bacterial membrane phospholipids containing phosphatidylethanolamine (PE) or phosphatidylglycerol (PG) headgroups with varying acyl chain lengths and unsaturation.

Compound	Neutral mass (Da)	m/z	Retention time (min)	Fold change BceAB-S/BceAB	Abundance (blank SEC buffer)	Abundance (BceAB)	Abundance (BceAB-S)
16:0,18:1-PE	726.5572709	744.5911	12.24583333	1	15434	137030175	169902003
	717.528819	718.5361	11.27378333	3	2322420	47944420	131385899
		744.5521	11.33056667	7	536775	18480519	126943918
18:1,18:1-PG	774.5388628	792.5727	10.96735	16	132148	5711045	93612435
16:1,18:1-PE		716.5204	10.99031667	9	931064	8174578	73047437
16:1,18:1-PG	748.5231464	766.557	10.91053333	4	321539	18392217	67275936
16:0,17:1-PE	703.5141443	704.5214	11.13876667	0	218482	155717937	61845693
17:1,18:1-PE		730.5366	11.21771667	2	108686	27382601	60622538
14:0,18:1-PE and 16:0,16:1-PE	689.4979708	690.5052	10.92211667	5	1272462	10534979	56748474
	662.444946	680.4788	11.45546667	1	30580363	65770820	39931063
16:1,18:1-PG	746.5082064	764.542	10.615	13	76134	3164362	39907909
mostly 17:1,18:1-PG, also seems to be some 19:1, 16:1-PG	760.5236264	778.5575	10.84218333	5	1204536	5277525	24490828
	1431.01688	1449.051	12.97536667	1	85367	22253515	22989316
16:0,17:1-PG	734.5080346	752.5419	10.78536667	1	4881959	28672159	20945868
	778.5339813	796.5678	12.07678333	5	4801	4467884	20336798
		1462.08	11.3191	12		1660643	19731747
16:0,16:1-PG	720.4922012	738.526	10.5576	3	1100881	6421011	18892758
	731.5449804	732.5523	11.50011667	0	1081469	55326525	18799739
	1404.999669	1423.033	12.9867	0	29872	34609931	13369987
	1442.031193	1488.097	11.3191	23	0	558191	13086462
	1402.985959	1421.02	12.8599	1	0	13811583	12930388
	596.5363145	614.5701	12.07678333	1	10244143	14620751	12838713
16:1,17:1-PE		702.505	10.8536	8	0	1540442	12403659
	1457.031506	1475.065	12.97536667	3	86301	3749796	11787983
		785.6169	12.24583333	1	88713	8459669	10909892
modified undecaprenyl phosphate	977.7223407	978.7296	11.95278333	0	409499	36776073	10107547
	624.5663668	607.5631	12.29158333	1	5838129	10438317	8979654
		1447.037	12.8599	4	0	2134872	8518405
2MH of 716.5204 (16:1,18:1-PE)		1432.033	11.01323333	15	0	549595	8261081
	745.5603042	746.5676	11.62488333	5	42095	1588685	8210364
16:0,19:1-PG and 17:0,18:1-PG	762.5390984	780.5729	11.13876667	1	1410439	12475241	8044466
16:1,16:1-PE	687.4824514	688.4897	10.615	16	51617	466462	7618148
	716.5515296	734.5854	12.75736667	1	0	7449241	7548178
adduct of m/z 689.49? (MS/MS spectrum of 1465 shows a 690.50 fragment ion)		1465.044	10.95568333	57	0	132049	7490548
	1376.970898	1395.005	12.87153333	0	1797905	35143235	7126827
	1012.765486	1030.799	12.34743333	10	672471	728210	7065010

Based on the results above it is clear that significant amounts of phospholipids co-purify with BceAB-S. However, unambiguous identification of the precise phospholipid(s) observed in the cryo-EM maps would require exceptionally high-resolution maps and rigidly bound phospholipids. For this reason we refrained from speculating on the exact identity of the lipids we observed.

In our opinion the densities observed between BceAB and BceS likely arise from co-purified membrane phospholipids rather than LMNG detergent. In our experience, when LMNG is observed in a cryo-EM map it is often bound quite rigidly and displays a defined “X” shape characteristic of the detergent with short 12-carbon acyl chains and two maltose headgroups. Below is a comparison of the densities observed in BceAB-S and an LMNG density recently observed in a cryo-EM reconstruction we performed on an unrelated and unpublished membrane protein complex at similar overall resolutions.

As can be seen from the comparison above, the phospholipids contain significantly longer acyl chain features (16-18 carbons versus 12 in LMNG), and LMNG displays strong features for the two maltose headgroups. Based on the lipidomic profiling and observed cryo-EM densities, we favor modeling of lipids rather than LMNG in the space that fills the gap between BceB and BceS TM helices.

We agree with the reviewer that understanding the lipid/detergent binding properties of BceAB-S is an important aspect of the overall structure and function of the complex. In the future we plan to perform detailed lipidomic analysis on these complexes after expression and purification from the native organism *Bacillus subtilis*. Such experiments require extensive controls and data analysis that is outside the scope of the current manuscript.

In the current manuscript we refrained from modeling a lipid in the UPP binding pocket that was identified in our previous *PNAS* paper. In the context of the BceAB-S structures in the current manuscript, there is indeed a “lipid-like” density in the UPP pocket in most of the reconstructions. However, compared to the previous reconstructions of BceAB alone, the map features for this lipid are significantly weaker and more fragmented in the maps obtained for the BceAB-S complex (see below). While we believe that this density likely arises from some partial occupancy of an undecaprenyl type lipid (which was also identified in the lipidomic profiling above), we refrained from modeling such a lipid into the current BceAB-S cryo-EM maps in order to avoid overinterpretation of weak and fragmented map features.

Minor comments:

The authors solved the structure of the complex solubilized in LMNG and use the word detergent “micelle” to describe the solvent seen around the hydrophobic region in the cryoEM reconstructions. I encourage the authors to use the word “belt” instead of micelle, as the micelle is a different object. It has been shown that LMNG micelles are long rod-shaped objects of around 400-600 monomers (aggregation number). With detergent quantification, we know that there are about 150 LMNG monomers around a 12TM type-IV ABC transporter, so I imagine there might be around 200 or less around the Bce complex. It is thus a different object than the micelle.

Thank you for the correction; we have made the change to describe a detergent “belt” rather than “micelle” throughout the text.

Figure S5: The authors mention a different orientation of TM helices of BceS. They have done an extensive analysis of cryoEM data, truly state-of-the-art, and a nice work to reach as high resolution as possible while looking at flexibility. The density maps look really good, and the model placing looks convincing. However, even at this resolution, it is still possible to make a mistake in the modelling so any help can be a good. I wonder about the QseC and ArcB, if by flipping the view they would look like NarQ ones, thus making BceS different on its own. Can the authors comment on alphafold predictions of BceS, and see if the helices are modelled the same way as they did. If yes, then it’s another point that this is correct. If no, it would be interesting to see the reverse modelling to judge what it looks like.

We thank the reviewer for this interesting comment. Indeed, the structures of QseC and ArcB that were determined with NMR at high temperature are difficult to interpret given that they are monomeric rather than dimeric. One could envision that a 180° rotation of these structures may give an orientation similar to NarQ. However, the monomeric state of

these structures limits their utility in this regard, and makes extensive comparison difficult. However, given the intersection angle between TM-1 and TM-2 observed in these NMR structures, we feel the configuration more closely resembles that of BceS.

In contrast to the situation with QseC and ArcB, we are quite confident in the orientation of helices we have built for BceS, as the map quality throughout both TM helices and the connecting loop between them is quite well resolved. As the reviewer suggests, AlphaFold predictions of BceS would lend support to the configuration that we have modeled. Not only have we modeled a dimer of BceS with AlphaFold Multimer V2 (see below), but we have also modeled the entire BceAB-S complex, and the predictions are remarkably similar to the experimentally determined complex. The AlphaFold predictions thus lend support not only to the modeling of BceS TM helix configurations, but also overall assembly of the BceAB-S complex. We look forward to incorporating these modeling results, and similar results on complexes from related bacterial species, into a forthcoming review on Bce modules.

“This result is consistent with our structural analyses demonstrating that nucleotide binding to BceA induces full closure of the NBD dimer in the isolated BceAB transporter¹², but not when the latter protein is in complex with BceS (Suppl. Fig. S9).” I believe the authors want to refer to SFigure 8 instead, I don’t have a SFigure 9.

Thank you for pointing out this error. This reference has been updated to Figure 6C, which compares the NBD dimer in the isolated BceAB vs the BceAB-S complex.

Figure 4 shows a beautiful example of protein flexibility sampled and analyzed by cryoEM, revealed by 3DVA analysis. The movements shown by this analysis (and also using 3D classification) are very intriguing and ask for many questions on the role of this variability on the kinase autophosphorylation function, to be addressed in further studies I hope. It would be useful to measure these movements (rotation, translation, etc...) more

accurately to analyze protein flexibility in terms of mechanics. Not required for this work, but perhaps in later studies.

We thank the reviewer for their interest in the flexibility revealed in our current study, and fully agree that the movements demonstrated are very intriguing. As the reviewer suggests, it is highly likely that these protein dynamics are intricately involved in enzymatic regulation and signaling throughout the complex. At this time, it is difficult to directly link specific conformational dynamics to the overall signaling mechanism. However, we look forward to further investigating the precise link between protein flexibility and autophosphorylation in future studies.

Reviewer #2 (Remarks to the Author):

George, et al. report the cryoEM structure of the BceAB-S complex from B. subtilis in both the apo and ATPγS-bound states. The structure reveals how an unusual ABC transporter, BceAB, interacts with BceS, a sensory histidine kinase. BceAB is quite unique among ABC transporters; though most similar to MacB and LolCDE, it differs substantially in its extracytoplasmic domains. The authors show that BceS binds to BceAB via the transmembrane region, and suggest that BceS activation may be triggered by conformational changes in the ABC transporter during the process of bacitracin detoxification. Comparing the structure of BceAB-S in the apo and ATPγS states reveals a kinked conformation of BceS, which may be involved in communication between the ABC transporter and the kinase domain. This communication was probed with a small number of mutations in an ATPase activity assay.

So overall, the cryoEM structures appear to be well-determined and well-refined. The structures presented are intriguing and provide our first insights into how this unprecedented combination of an ABC transporter and sensory histidine kinase work together to sense antibiotics targeting the cell wall. We think this work will be of interest to several scientific communities, including researchers interested in ABC transporters, cryoEM of membrane proteins, antibiotic resistance, and bacterial mechanisms of environmental sensing and response. While we include some questions and suggestions for revision below, overall we thought this was a very exciting story and have only relatively minor criticisms.

We thank the reviewers for their very supportive comments.

MAJOR COMMENTS:

No Major Comments

MINOR COMMENTS:

Did the authors state what “bce” stands for? We couldn’t find it.

The Bce modules are named after the BceABRS module from *B. subtilis*; this connection has been clarified in the manuscript by adding the following:

“Of all such modules, the BceAB-RS system from *Bacillus subtilis*, after which the modules have been named, has been most extensively characterized⁸⁻¹⁰.”

“Bce” in the BceABRS module stands for “bacitracin efflux,” as it was initially assumed that the module provided resistance by exporting bacitracin from the cell (Ohki R. et al. (2003) Molecular Microbiology). However, as later studies identified that the BceAB transporter does not have the ability to move bacitracin across the membrane, we have chosen to not elaborate on the name in the manuscript to avoid confusion over the function of the module.

It would be nice to show a model figure for how the authors think the whole system functions (either at the beginning by way of introduction, or at the end to summarize).

We have added Supplemental Figure S9 to show a cartoon model summarizing structural data from the current manuscript, as well as information obtained from previous studies in the literature. In addition, we expanded the discussion to reference this new figure and summarize the overall proposed signaling mechanism.

It would be helpful to add lines or shading indicating where the membrane region is on all figures (as in Fig. 2); this is missing in several places (e.g., Fig. 3). Also labels for cytoplasm, extracellular, etc.

We have added dashed lines and “extracellular” or “cytosolic” labels in all appropriate places in figures and figure legends. Thank you for pointing out this simple addition that we agree provides more clarity.

Line 114: "two TM helix states": This wording is a little confusing to me. I take it to mean two different conformations that differ in the transmembrane region, but I am not completely sure this is correct.

This interpretation is correct. The wording has been updated to “two TM helix conformations” for clarity.

Line 115: "two monomers": this makes me picture the BceS protomers as binding independently to BceAB, but it looks like BceS forms a dimer. Say that more clearly?

This sentence has been updated to “the BceAB transporter is complexed with the BceS dimer” for clarity.

For readers less familiar with histidine kinases, it would be helpful and enlightening to compare the structure of BceS with other available structures. e.g., are the kinase domains similarly oriented? Are stalk lengths similar? Etc...

We appreciate the reviewers comment, and agree that it would be helpful to compare the structure of BceS with other available histidine kinase structures. We attempted to make this comparison, at least in the transmembrane region in Supplemental Figure S5. While we agree that a more thorough comparison to all available histidine kinase structures is warranted, we are also currently at the limit of space available in a single manuscript. At this time we prefer to save such detailed comparisons across a large family of available structures for a forthcoming review that is being prepared.

Fig. 3A,B: The lipid mediated interaction between BceAB and BceS is quite interesting. However, the details of how this interaction is stabilized is not particularly clear in Fig. 3. Also, what sorts of direct protein-protein interactions are made at this interface? It might be helpful to add additional panels to this figure showing how each protein interacts with the bridging lipid and how BceS interacts with BceB.

We have significantly expanded Figure 3 to include three new panels (C-E) showing the details of interaction between protein and lipid, as well as the protein-protein interactions between BceB-BceS. These new panels have been referenced at the appropriate locations within the main text.

Fig. 4B: The movie showing the 3DVA is nice, and Fig. S6F nicely illustrates one kind of motion in BceS. In contrast, it is a bit more difficult to see what is moving relative to what in Fig. 4B. Could these motions be shown in a different way (perhaps like the model-based representation in Fig. S6F)?

We agree that the representation in Fig. S6F nicely displays the motion in BceS, and it may be a bit more difficult to conceptualize this motion from Fig. 4B. Unfortunately, a figure such as Fig. S6F requires an atomic model to separately be refined into individual cryo-EM maps. While we could attempt low-resolution refinement of atomic models into intermediate maps produced during 3D variability analysis, or individual maps obtained through 3D classification, in practice the density for BceS in these maps is fragmented and weak in most maps. Thus, atomic model refinement into such maps is non-trivial and would surely suffer from a lack of experimental data in the most mobile regions. For this reason, we refrained from such a model-based approach for figure generation, and chose rather to show the raw cryo-EM maps produced through 3D classification.

Line 268: refers to Suppl. Fig. S9, but I don't see that figure in the reviewer copy.

Thank you for pointing out this error. This reference has been updated to Figure 6C, which compares the NBD dimer in the isolated BceAB vs the BceAB-S complex.

In a couple of places, the authors refer to “tilted BceS”. Is this the same as “kinked”? If so, using the same nomenclature throughout would be more clear.

Thank you for pointing out this discrepancy. We have made the appropriate changes to only refer to the altered configuration as “kinked”.

Fig 6A: ATP not labeled or mentioned in figure legend.

We have added labels to the figure, and updated the figure legend to indicate that the nucleotides are shown as magenta spheres.

Table S1: Include model-map CC values.

Thank you for pointing this out. We have added the values to table 1.

Line 135-143: This seems to me like a surprising observation that would be interesting to comment on further. Is it possible that some of the structures are incorrect? Are these regions thought to derive from a common ancestor for both the left and right handed turns? In addition to the turn, it looks like the helix packing angles / superhelix handedness might also differ. Is this convergent evolution? Because the different connectivity between structures is surprising, it would be good to show density for the BceS TM1-TM2 linkers in Fig. S5 (one could imagine showing the linker density for NarQ and HtrII as well, for completeness).

Please see response to reviewer #1 above regarding this matter. Indeed, the NMR structures of QseC and ArcB were determined with NMR at high temperature, and are monomeric rather than dimeric. Thus, interpretation of these structures is difficult and is currently of limited utility in understanding the overall packing of transmembrane helices across the entire family of histidine kinases. However, as elaborated upon above in response to reviewer #1, we are quite confident of the overall transmembrane helix configuration in BceS. In addition to the AlphaFold model shown above, we have also included an inset panel in Fig. S5C to show the TM1-TM2 linker that is well supported by the experimental cryo-EM map. From these models it is clear that BceS has the opposite TM helix configuration of NarQ. As there are currently very few high-resolution structures of the TM domain of histidine kinases, it is difficult to speculate whether such differences/similarities can be attributed to ancestral lineages and/or convergent evolution. Further high-resolution structures and extensive molecular modelling of kinases from various organisms will be required to support or refute such lineages.

Line 252-258: Might the kinking and movement of BceS be caused by nucleotide binding to the CA domains rather than nucleotide binding to BceA? If the interactions between the BceS and BceB TM regions don't change much, how would the nucleotide state of BceA be transmitted to BceS to cause the observed conformational change? This part of the mechanism is not clear to us, though perhaps it remains to be unraveled.

We thank the reviewer for this very important question. While the resolution of the maps in the CA domain is rather low, we do not believe that nucleotide binding to BceS is inducing the observed conformational change. Even at the modest resolutions obtained in this domain, we do not see any faint hints of density that would indicate nucleotide binding in the CA domain. Moreover, it is clear that even in the ATP γ S bound structure the orientation of the CA domains relative to the DhP bundle does not change, despite the kink that is observed beneath the HAMP domain. Thus, it appears that BceS is still far from adopting a conformation that represents the Michaelis complex which would precede the autophosphorylation reaction. It is important to note that the ATP γ S bound state shown in Fig. 5A represents ~25% of the total particle population (Suppl. Fig. S7A). The remainder of the particles in this dataset still maintain a high degree of flexibility in the BceS soluble domains. Thus, there are likely many intermediates that remain to be resolved in the overall cycle of kinase activation. The precise mechanism whereby BceS “senses” changes in the transmembrane and/or cytosolic regions of BceAB remains slightly murky. Most importantly, how binding of bacitracin to the extracellular region aids in triggering these overall changes and kinase activation remains obscure. We look forward to teasing out these mechanistic details in future work.

Regarding the ATPase Assay, does BceS have ATPase activity on its own? We were a little uncertain about how to interpret the results of the ATPase assays in light of the possibility (?) that BceA and BceS may both be capable of hydrolyzing ATP, and could also potentially allosterically regulate each other's activity.

In our hands, purified BceS alone (lacking BceAB) is devoid of any *in-vitro* ATPase activity (data not shown). This mirrors the situation observed *in-vivo* in previous literature reports, where BceS is unable to undergo autophosphorylation in the absence of functional BceAB. More importantly, we have tried quite extensively to reconstitute *in-vitro* autophosphorylation of BceS using the detergent solubilized and purified BceAB-S complex. Based on extensive phos-tag gel and phos-tag fluorescent stain assays, we have yet to detect any *in-vitro* phosphorylation of BceS using the detergent purified complexes. This lack of autophosphorylation likely reflects the missing component of bacitracin-UPP interaction to initiate signaling and activation through the overall complex. Thus, we are quite confident that the ATPase activity in the current manuscript arises almost entirely from BceAB, rather than a combination of BceS mediated turnover.

The differences between the previous structure of ATP-bound E-to-Q mutant of BceAB, vs the current structure of BceAB-S bound to ATP-gamma-S are generally interpreted in terms of differences between BceAB in isolation and BceAB bound to BceS. But sometimes different strategies to stabilize the ATP bound state can give differing results. In addition, given the presence of multiple non-equivalent ATP binding sites (two different sites in BceA, plus 2 sites in BceS), the situation is extra complicated. I might suggest being a little more conservative in how these structural differences are interpreted, or

perhaps mutants could be made to probe the role of ATP-binding at various sites in BceA vs BceS?

We appreciate the reviewer's comments and agree that sometimes different strategies of stabilizing the nucleotide-bound state may generate differing results. For this reason we tried initially to stabilize the nucleotide-bound BceAB-S complex using a similar E-to-Q approach that was used in our initial publication of BceAB alone. For reasons that still remain unclear to us, the BceAB(E169Q)-BceS complex is unstable during purification, and we often observed dissociation of BceAB(E169Q) from BceS during affinity and gel-filtration chromatography (even in the absence of added nucleotide). Thus, in order to observe the effect of nucleotide binding on the BceAB-S complex we turned to the nucleotide analog ATP γ S. For now this approach provides the most accessible means to observe the effect of nucleotide binding on the overall BceAB-S complex, but we acknowledge that it is not without potential caveats. Future biochemical and structural work will be aimed at unraveling the precise role of each potential ATP binding site (two in BceA plus two in BceS) in overall complex dynamics and function. Such mutational analysis is non-trivial, as extensive construct design and validation will be required in order to assemble complexes in which the mutations only arise in the desired location of one out of two identical protein subunits.

Reviewer #3 (Remarks to the Author):

The manuscript by Natasha George and Benjamin Orlando describes novel structures of the B. subtilis BceAB-S complex, implicated in antimicrobial peptide sensing (such as bacitracin). The manuscript builds on previous studies by the same group on the BceAB complex alone, which is an ABC transporter system evolved to sense, rather than transport, certain ligands. The transporter interacts with a two component system containing a His kinase (BceS) and response regulator. The current manuscript describes cryo EM structures of the BceAB transporter in complex with a BceS dimer. Somehow, bacitracin binding by BceAB activates BceS, leading to autophosphorylation of BceS' DHp domain. Although compromised by conformationally flexible particles, the new data describe apo and ATP γ S-bound structures of the BceAB-S complex and reveal conformational changes of BceS that may be important for signal transduction.

Major points

- For a general reader, it would be useful to illustrate the signal process as a cartoon in Fig. 1.

Thank you for pointing out this important point for a general audience. We have updated Fig. 1A to include the aspects of the overall signaling process. Additionally, in response to the comment from reviewer #2 above, we have also included the additional Supplementary Fig. S9. This figure demonstrates a schematic representation of the overall signaling process compiling information gleaned from our current studies as well as previous biochemical assays in the literature. We hope that the modification of these two

figures simplifies the complicated process of signaling through the BceAB-S complex for a more generalized audience.

- Fig. 3 – interaction of BceAB with BceS: The interaction of the BceS dimer with BceAB is not described in detail. Perhaps this is due to limiting resolution of the interface. However, are there any conserved residues that cluster at the interface? Considering the small interface, it is surprising that a stable complex is formed. In addition, the density in the lipid cleft could be shown in more detail (perhaps as a supplemental figure) to support the interpretation as a lipid molecule. Could it be a detergent molecule?

We have significantly updated Figure 3 by adding three new panels detailing the protein-lipid and protein-protein interactions between BceB-BceS. Please see response to reviewer #1 above for a discussion about co-purified lipids and interpretation of lipid densities and detergents, as well as AlphaFold modelling supporting the overall assembly of the BceAB-S complex. Indeed, the interface between BceB-BceS is quite small and largely hydrophobic in nature. Currently, from individual sequence analysis and structural interpretation, there are not obvious individual conserved residues that stand out to us as being critical for this interaction. However, deep co-evolution analysis may provide further insight into pairs of residues at this interface that have co-evolved to support interaction.

- Fig. 3 – BceS – BceA interaction: I understand that BceS' DHp and CA domains are insufficiently resolved in most maps. However, Fig. 3A shows a complete model that suggests potential interactions of BceA with one BceS protomer. Could the authors speculate on whether or not this interaction is significant? Are there any conserved residues at the interface that may transfer conformational changes of BceA to BceS?

Indeed, the models do suggest potential interaction between the BceS HAMP domain/stalk helices and the outside face of one BceA protomer (particularly the beta sheets that form the A-loop which contains a conserved Tyr-13 that pi-stacks with the adenine ring of ATP). However, this potential interaction is likely weak as the maps in this region still suggest a high degree of flexibility and conformational variation. At this time, it is difficult to speculate on the degree of significance of these potential interactions in the overall signaling mechanism. Previous reports (Koh, A. et al. *Mol. Microbiology* (2021)) have demonstrated that BceS activation is accompanied by piston-like movements of the BceS TM helices, suggesting that signaling originates from conformational coupling to transitions in BceB TM helices. Whether or not BceA-BceS interactions also play a role in this process will require extensive future mutational studies, using an *in-vivo* signaling assay to probe the role of specific protein-protein interfaces in the overall signaling mechanism.

- ATPgS-bound state of BceAB: This structure reveals a closer association of BceA's nucleotide binding domains in the presence of ATPgS. Yet, as stated, the NBDs do not close completely. This is in contrast to a previously obtained structure of a Walker-B mutant of BceAB bound to ATP. Thus, is it possible that the new structure is not completely

closed due to ATP γ S? Have the authors tried to stabilize the ATP-bound BceAB-S complex by mutagenesis, as described in Ref. 12? This seems important because the current structure would suggest that perhaps bacitracin sensing, i.e. changes within the TM region, are necessary for complete NBD closure.

This is an excellent point. Please see the detailed response to reviewer #2 above regarding the differences between ATP γ S and the E169Q mutation. In brief, we tried extensively to utilize the E169Q variant to trap a nucleotide bound structure of the entire BceAB-S complex. For reasons that are still unclear to us, the interaction between BceAB(E169Q) and BceS is unstable compared to the wild-type complex (even in the absence of added exogenous nucleotide). While it is possible that the differences we observe between these two strategies are partially due to the identity of the nucleotide used to trap the complexes, we favor a model in which bacitracin sensing facilitates further NBD closure in the entire BceAB-S complex. The fact that the ATPase activity of BceAB-S is reduced compared to isolated BceAB suggests that additional factors (ie: bacitracin sensing) are required to induce full NBD closure and maximal ATP hydrolysis. Such a model also lends support to the notion that BceAB-S would not undergo futile rounds of ATP hydrolysis (and uncontrolled autophosphorylation and signaling) in the absence of the critical component bacitracin.

- Fig. 5 – bending of BceS: The ATP γ S-induced bending of BceS is interesting, however, it is not described how the kink is induced. What conformational changes of BceA reposition the BceS protomer?

This is a great question that follows the line of reasoning from two points above. As the overall transmembrane helix configuration does not change significantly in response to ATP γ S binding, it seems logical that the kink in BceS is primarily a result of altered BceA configuration. However, it is important to note that in the dataset with ATP γ S present, ~25% of the particles adopt the kinked BceS configuration, whereas ~50% of the particles still display largely disordered BceS cytoplasmic domains. In the particle population where BceS remains largely disordered, the BceA domains adopt the same configuration (ATP γ S bound) as seen in the particle population with kinked BceS. Thus, at least from our *in-vitro* cryo-EM analysis it appears that kinking of BceS and movement of BceA are not intimately associated with one another 100% of the time. The significance of interaction and conformational coupling between BceA and BceS is something we look forward to examining with detailed *in-vivo* assays in the near future, but is outside the scope of this current manuscript.

- Fig. 5 - ATPase assays: It would be very interesting to compare the hydrolytic activity of BceAB w/o BceS in the presence and absence of bacitracin/UPP.

We agree that BceAB activity with and without bacitracin/UPP would be interesting to investigate and important for a complete understanding of the function of the module. However, as described in our previous publication, this assay has several challenges.

BceAB co-purifies from *E. coli* with a lipid that differs from UPP in the replacement of the terminal phosphate with amino-arabinose. This lipid is highly abundant in *E. coli* and likely co-purifies with BceAB due to its overall similarity to UPP, but unfortunately the amino-arabinose headgroup interferes with bacitracin binding. In addition, bacitracin is most active in complex with Zn^{2+} . We previously found that Zn^{2+} can inhibit BceAB ATPase activity on its own, likely by replacing Mg^{2+} in the ATP binding site. These barriers make this assay unfeasible with our current expression and purification strategies. Extensive development in expression and purification protocols will be required to extract BceAB from native *B. subtilis* and hopefully co-purify the complex with the native UPP lipid.

- Discussion: The discussion lacks a (speculative) description of the overall signaling process. How does bacitracin sensing lead to BceS autophosphorylation? What kind of conformational changes of BceAB are expected that could activate BceS? Where does bacitracin/UPP bind and would that interfere with BceS binding?

We have added an additional Supplemental Figure S9 depicting a speculative model of the overall signaling process that incorporates features observed in our cryo-EM analysis as well as previous biochemical assays in the literature. We have expanded the discussion to include a paragraph describing this overall process, and the hypothetical steps involved in signaling. It should be noted that many of the intricate details of bacitracin sensing and signaling through the entire complex remain obscure. However, the current model in Figure S9 compiles the strongest information currently available.

The extracellular domain of BceB is known to be involved in bacitracin sensing, and our previous structural analysis of BceAB suggests that UPP-bacitracin complexes would be positioned to place bacitracin just within interaction range of the extracellular domain. Precisely how the extracellular domain recognizes UPP-bacitracin complexes awaits high-resolution structural analysis of a substrate engaged complex. However, it seems clear that UPP-bacitracin binding would not interfere with the interaction between BceAB-BceS. Rather, we envision that the conformational changes induced by UPP-bacitracin recognition and concomitant ATP hydrolysis in BceAB would be transmitted through BceS starting at the transmembrane helices, and propagating downward through the cytoplasmic domains.

Minor points

The main text includes detailed experimental information that could be moved to the supplement, should space be limiting.

Reviewer comments, second round

Reviewer #1 (Remarks to the Author):

The authors have answered all my concerns, and their answers make good sense. I recommend this article for publication.

Reviewer #2 (Remarks to the Author):

The authors have adequately responded to all of my comments.

Reviewer #3 (Remarks to the Author):

The authors have addressed all of my questions and concerns. I do not have any additional comments.

[redacted] We are very pleased that all three reviewers found our previous responses to their comments satisfactory. Highlighted below are point-by-point responses to each reviewer. The original comments and suggestions from the reviewers are highlighted in *italicized and indented text*. Our responses are **highlighted in bold beneath each reviewer critique**.

Reviewer #1:

The authors have answered all my concerns, and their answers make good sense. I recommend this article for publication.

We thank reviewer #1 for their help in reviewing our manuscript and recommending for publication.

Reviewer #2:

The authors have adequately responded to all of my comments.

We thank reviewer #2 for their help in reviewing our manuscript.

Reviewer #3:

The authors have addressed all of my questions and concerns. I do not have any additional comments.

We thank reviewer #3 for their help in reviewing our manuscript.